# Characteristics of Spiral Patterns Formed by Coaxial Interference between Two Vortex Beams with Different Radii of Wavefront Curvatures

**Jingbo Ma [1], Peng Li [1,2,3,*] and Yuzong Gu [1,2,3]**

[1] Research Center for Physics of 2D Opto-Electronic Materials and Devices, School of Physics and Electronics, Henan University, Kaifeng 475004, China; 17352576816@sina.cn (J.M.); yzgu@vip.henu.edu.cn (Y.G.)
[2] International Joint Research Laboratory of New Energy Materials and Devices of Henan Province, Kaifeng 475004, China
[3] Institute of Micro/Nano Photonic Materials and Applications, School of Physics and Electronics, Henan University, Kaifeng 475004, China
* Correspondence: lilipengpeng@vip.henu.edu.cn

**Abstract:** Spiral pattern is formed for coaxial interference between two vortex beams with different radii of wavefront curvatures and different topological charges (TCs). A theoretical model considering various parameters (such as phase difference, radius of wavefront curvature, and TCs) is established to predict all kinds of interference patterns. An improved Mach-Zehnder interferometer is set up in an experiment to generate different kinds of spiral patterns and verify the theoretical model. The number of spiral lobes is determined by the absolute value of TCs' difference between two vortex beams, and the twist direction relates to the sign of TCs' difference and the difference of reciprocals for the radii of wavefront curvature, clockwise for the same sign, and counterclockwise for the opposite signs. The twist direction of the spiral pattern reverses and the lobes direction near the core of the pattern changes obviously when the spherical wave changes from convergence to divergence.

**Keywords:** Laguerre-Gaussian vortex beam; topological charge; radius of wavefront curvature; coaxial interference; Mach-Zehnder interferometer; spiral pattern

## 1. Introduction

The Laguerre-Gaussian (LG) vortex beam with an $\exp(il\varphi)$ azimuthal phase, where $l$ is called topological charge (TC), has a doughnut intensity distribution and carries an orbital angular momentum (OAM) of $l\hbar$ ($\hbar$ is the reduced Planck's constant) per photon [1]. It is widely applied in realms such as optical communications [2–4], trapping, and manipulating micrometer-sized particles [5,6], measuring the rotational speed of a rotating object [7], fabricating chiral nanostructures [8–10], and atomic optics and spectroscopy [11], etc. The phase structure of a vortex beam presents $l$ intertwined helical phase fronts. According to the sign of TC $l$, the vortex beam can be divided into two types, the left-hand vortex beam (positive $l$) and the right-hand vortex beam (negative $l$). Although the phase changes in different directions (counterclockwise and clockwise) and surround the singularity for left- and right-hand vortex beams with the same TC value and the opposite sign, their beam patterns show the same annulus structure, and we cannot distinguish them from each other simply by the beam pattern [12]. The determination of TC carried by the vortex beam has become a popular research topic in recent years, and various techniques based on the interference and diffraction characteristics of vortex beams have been proposed [12–38]. The most commonly used TC detection method is based on Mach-Zehnder interferometer (MZI) [19–24], which uses beam splitters in combination with mirrors to interfere two beams passing through two arms of different paths, and allows their phase structure to be compared with each other. For off-axis interference between two plane wave vortex beams of different TCs, fork-shaped fringes form, and the sign and magnitude of TCs' difference

can be determined by analyzing these interferograms and the orientation between two vortex beams [13–21]. For coaxial interference between two plane wave vortex beams of different TCs, a petal-shaped pattern forms, and the petal number is just the magnitude of TCs' difference [23–27]. For coaxial interference between a plane wave vortex beam and a spherical wave beam, a spiral pattern forms, and it can be used to decide both the sign and value of TCs [20,28–32]. Furthermore, the coaxial interference between a plane wave vortex beam and a spherical beam also has significance application in fields such as optical measurement [39] and the fabrication of chiral microstructures materials [9].

The coaxial interference of a wave possessing helical wavefront (vortex beam carrying TC of *l*) with a reference wave of spherical wavefront (tightly focused $TEM_{00}$ Gaussian beam) produces the spiral pattern, exhibiting $|l|$ fringes radiating from the center. The number of spiral lobes reveals the value of TC, and the rotation direction of the spiral fringe (twist direction) is determined by the sign of *l* and relative wavefronts curvature radii of two beams. When the plane wave vortex beam is interfered by the copropagating divergent spherical beam, the twist direction is clockwise for negative *l* and counterclockwise for positive *l*. On the other hand, the opposite twist direction can be obtained when the plane wave vortex beam is interfered with the copropagating converging spherical beam. M. S. Soskin and coworkers studied the interference characteristics between the screw-dislocation wave (vortex beam) and spherical wave systematically for the first time [28–30]. They showed a simple equation to describe the interference pattern produced by the usual plane wave and the screw-dislocation wave (vortex wave) when both waves are propagated in the same direction [28], and established important rules for the determination of the TC value and helicity of screw-dislocations [29]. They also tested the spatial structure of optical vortex helical wavefronts by using various interference arrangements and precise measuring techniques and demonstrated that coaxial interference pattern contains the data both on the phase difference of the reference and tested beams at each point of the observation plane, as the parameters of the reference wave (such as amplitude distribution and radius of curvature of the wavefront) are known [30]. In 2016, J. Jin et al. demonstrated a metasurface composed of two sets of elliptical nanoholes to generate and detect the OAM [32]. An array of elliptical nanoholes allow converting the circularly polarized light into the cross-polarized vortex beam, and another elliptical nanoholes array allow generating the spherical reference wave to interference with the vortex beam coaxially and detect the OAM by the spiral pattern. The twist direction of spiral fringes for the interference between the helical wavefront and the divergent wavefront is opposite to the case of convergence wavefront interference [32]. Recent work by D. Yang et al. proposed a new method for the radius of curvature of spherical wave measurement based on the vortex beam interference and Fermat's spiral fitting [40]. The interference model during theoretical analysis considers the interference between the plane wave vortex light and the spherical wave to describe the characteristics of spiral fringe, while in the actual experiment the interference between a collimated vortex light and a convergent (or divergent) fundamental Gaussian beam (tightly focused $TEM_{00}$ Gaussian beam) is used [32]. There exists an obvious difference between the spherical wave (the simply light field expression is $E = (A/r) \exp[ikr^2/(2R)] \exp[ikz + i\varphi_1]$, where *R* denotes the radius of wavefront curvature of the spherical wave) and the fundamental Gaussian beam (the simply light field expression is $E = [A_2/\sqrt{1 + (z/z_R)^2}] \exp[-r^2/w^2] \exp[ikr^2/(2R)] \exp[ikz + i\varphi_2]$, where $z_R$ is Rayleigh length and *w* is the beam waist, and suppose that the minimum waist is located at *z* = 0). This difference will lead to the deviation of experimental results from the simulation ones. As far as we know, there is no quantitative study on the coaxial interference between two actual vortex beams with different radii of wavefront curvature. The aim of this article is to study the factors affecting the spiral pattern formed by coaxial interference between two vortex beams with different radii of wavefront curvatures, both theoretically and in experiment, and give a general rule for the determination of TCs carried by an unknown vortex beam. The novelties are the establishment of a theoretical model describing the coaxial interference of actual $LG_{0l}$ vortex beams, and the explanations why

the twist direction reverses and why the lobes direction near the core of the pattern changes obviously when the spherical wave changes from convergence to divergence.

In this paper, a systematic study on coaxial interference patterns between two vortex beams with different radii of wavefront curvature is presented. Section 2 is devoted to comprehensive description of the theoretical model about coaxial interference between two vortex beams with different radii of wavefront curvature. Experimental procedures to generate the coaxial interference patterns between two vortex beams of different TCs are presented in Section 3. Experimental results and simulations, including effects of phase difference, radii of wavefront curvature difference, and TCs difference between two interference beams on the interference pattern, are discussed in Section 4. The article ends with the conclusion in Section 5. These results not only broaden the understanding of the interference between two vortex beams, but also provide guidelines for vortex beam-based metrology and vortex beam diagnostics.

## 2. Theoretical Analysis

We use the LG vortex beam as an example of optical vortex during our study. In cylindrical coordinates, the complex amplitude of the optical field for LG vortex beam $LG_{0l}$ carrying TC $l$ and oriented along $z$ is expressed as [21]:

$$E_{0l}(r, \varphi, z) = \sqrt{\frac{P_l(|l|+1)}{\pi n c \varepsilon_0 |l|! w_{0l}^2(z)}} \left[ \frac{\sqrt{2(|l|+1)}r}{w_{0l}(z)} \right]^{|l|} \exp\left[ -\frac{(|l|+1)r^2}{w_{0l}^2(z)} \right] \exp[i\psi_{0l}(r, \varphi, z)] \tag{1}$$

Here, $P_l$ represents the power of the vortex beam, $n$ is the refractive index of the transmission medium for light at wavelength $\lambda$, $\varepsilon_0$ is the permittivity of vacuum, and $c$ is the speed of light in vacuum. $w_{0l}(z) = w_{0l}\sqrt{1 + [(z - z_l)/z_{R_l}]^2}$ and $w_{0l}$ are the spot radius and the waist radius of $LG_{0l}$ vortex beam, respectively. $z_l$ and $z_{R_l} = \pi n w_{0l}^2 / [\lambda(|l|+1)]$ are the beam waist's position and Rayleigh length, respectively. The phase $\psi_{0l}(r, \varphi, z)$ is given by

$$\psi_{0l}(r, \varphi, z) = -(|l|+1)\arctan[(z - z_l)/z_{R_l}] + kr^2/[2R_l(z)] + l\varphi + kz \tag{2}$$

Here, $R_l(z) = z_{R_l}[(z - z_l)/z_{R_l} + z_{R_l}/(z - z_l)]$ is the radius of wavefront curvature, and $k = 2\pi n/\lambda$ denotes the wave vector modulus.

To generate the spiral pattern, two LG vortex beams carrying TCs of $l_1$ and $l_2$ are arranged to interfere coaxially. Here we consider $|l_1| \leq |l_2|$ for simplicity. When both two vortex beams propagate in the same direction, they are superimposed to be:

$$E_{total}(x, y, z) = E_{0l_1}(x, y, z) + E_{0l_2}(x, y, z)\exp(i\delta) \tag{3}$$

The interference pattern is governed by the interference light intensity distribution on a screen ($z$ constant):

$$I(r, \varphi, z) = 2nc\varepsilon_0 |E_{total}(r, \varphi, z)|^2 = 2nc\varepsilon_0 |E_{0l_1}(r, \varphi, z) + E_{0l_2}(r, \varphi, z)\exp(i\delta)|^2$$

$$= \frac{2P_1(|l_1|+1)}{\pi |l_1|! w_{0l_1}^2(z)} \left[ \frac{2(|l_1|+1)r^2}{w_{0l_1}^2(z)} \right]^{|l_1|} \exp\left[ -\frac{2(|l_1|+1)r^2}{w_{0l_1}^2(z)} \right] * \left\{ 1 + Q^2(r) + 2Q(r)\cos\left[ \psi_{0l_2}(r, \varphi, z) - \psi_{0l_1}(r, \varphi, z) + \delta \right] \right\} \tag{4}$$

Here $\delta$ is the phase difference between two vortex beams. $Q(r)$ is defined as

$$Q(r) = \sqrt{\frac{(|l_2|+1)^{|l_2|+1}|l_1|!P_2}{(|l_1|+1)^{|l_1|+1}|l_2|!P_1} \frac{w_{0l_1}^{|l_1|+1}(z)}{w_{0l_2}^{|l_2|+1}(z)} \left(\sqrt{2}r\right)^{|l_2|-|l_1|} \exp\left[ -\frac{(|l_2|+1)w_{0l_1}^2(z) - (|l_1|+1)w_{0l_2}^2(z)}{w_{0l_1}^2(z) * w_{0l_2}^2(z)} r^2 \right]} \tag{5}$$

The spiral pattern is determined by the interference term $\cos\left[ \psi_{0l_2}(r, \varphi, z) - \psi_{0l_1}(r, \varphi, z) + \delta \right]$. Here, $\psi_{0l_2}(r, \varphi, z) - \psi_{0l_1}(r, \varphi, z)$ can be specific expressed as

$$\psi_{0l_2}(r,\varphi,z) - \psi_{0l_1}(r,\varphi,z) = (|l_1|+1)\arctan[(z-z_{l_1})/z_{R_{l_1}}] - (|l_2|+1)\arctan[(z-z_{l_2})/z_{R_{l_2}}]$$
$$+[1/R_{l_2}(z) - 1/R_{l_1}(z)]kr^2/2 + (l_2-l_1)\varphi \tag{6}$$

The interference term on the detector can be further expressed as:

$$\cos[\psi_{0l_2}(r,\varphi,z) - \psi_{0l_1}(r,\varphi,z)+\delta] = \cos\left\{[1/R_{l_2}(z) - 1/R_{l_1}(z)]kr^2/2 + (l_2-l_1)\varphi + \delta'\right\} \tag{7}$$

Here, $\delta' = (|l_1|+1)\arctan[(z-z_{l_1})/z_{R_{l_1}}] - (|l_2|+1)\arctan[(z-z_{l_2})/z_{R_{l_2}}] + \delta$ is the effective phase difference. The equation for the maximum distribution of the interference light intensity (the bright fringe) will have the form:

$$[1/R_{l_2}(z) - 1/R_{l_1}(z)]kr^2/2 + (l_2-l_1)\varphi + \delta' = 2m\pi, \ (m=0,\pm1,\pm2\cdots) \tag{8}$$

Here, $m$ relates to the number of cycles in the interferogram. The curve plotted by the maximum light intensity governed by Equation (8) represents a Fermat's spiral [40]. The Fermat's spiral relates to both $1/R_{l_2}(z) - 1/R_{l_1}(z)$ and the difference of TCs $l_2 - l_1$. At the beam center ($r = 0$), the interference term shown by Equation (8) becomes $\cos[(l_2-l_1)\varphi+\delta']$, thus resulting in the phenomenon that $|l_2-l_1|$ fringes are born or vanished from the beam center. At other regions, with the increase of $r$ from 0, the interference pattern presents as a spiral pattern for $R_{l_1}(z) \neq R_{l_2}(z)$. The number of spiral lobes is just $|l_2-l_1|$. Equation (8) also indicates that the smaller the difference of reciprocals for wavefront curvature radii $1/R_{l_2}(z) - 1/R_{l_1}(z)$, the larger the radical coordinate $r$ under the same $\varphi$, and the smaller the number of cycles ($m$) in the interferogram. Further, the larger the number of cycles ($m$) in the interferogram, the denser the spiral will be. When $\varphi$ is fixed, bright and dark fringes appear alternately with the increase of $r$. The space $\Delta r$ between adjacent fringes satisfies $[1/R_{l_2}(z) - 1/R_{l_1}(z)]k(r+\Delta r/2)\Delta r = 2\pi$.

Compare the following two cases. (i) Fix the wavefront curvature radii $R_l(z)$ and exchange the TCs of two vortex beams, the interference term shown in Equation (7) becomes $\cos\left\{[1/R_{l_2}(z) - 1/R_{l_1}(z)]kr^2/2 + (l_1-l_2)\varphi + \delta'\right\}$; (ii) Fix the TCs and exchange the wavefront curvatures $R_l(z)$ of two vortex beams, the interference term becomes $\cos\left\{[1/R_{l_2}(z) - 1/R_{l_1}(z)]kr^2/2 + (l_1-l_2)\varphi - \delta'\right\}$. Except for the effective phase difference $\pm\delta'$, the interferograms for the two cases above are similar. We conclude that the twist direction of the interferograms between two vortex beams depends on both the sign of TCs' difference ($l_2 - l_1$) and the difference of reciprocals for wavefront curvature radii $[1/R_{l_2}(z) - 1/R_{l_1}(z)]$. If $1/R_{l_2}(z) > 1/R_{l_1}(z)$, the twist direction is clockwise for $l_2 > l_1$, while counterclockwise for $l_2 < l_1$. If $1/R_{l_2}(z) < 1/R_{l_1}(z)$, the twist direction is clockwise for $l_2 < l_1$ and vice versa.

## 3. Experiment Setup

The schematic of experimental setup is similar to our previous work [21] and shown in Figure 1. Firstly, the continuous-wave, single-frequency (line width of <100 kHz) 1064 nm laser from the collimator of Yb-fiber laser and amplifier (NKT Photonics, Koheras Y10) passes through the optical isolator and power-control system, which is composed of a half-wave plate (HWP) and a polarizing beam splitter (PBS). Then, the transmitted laser beam is recollimated again by using a plano-convex lens $f_0$ with a focal length of 500 mm. After that, the collimated laser beam is used as the fundamental Gaussian beam in the following experiment. Secondly, two spiral phase plates (SPPs, UPO Labs) of winding number corresponding to vortex orders, $l = 1$ and 2, allow converting the Gaussian beam ($|l=0\rangle$) to the vortex beam of $|l = \pm1, \pm2, \pm3\rangle$ [12,41,42]. A dove prism is used to convert the vortex beam of $|l\rangle$ to $|-l\rangle$. The MZI, composed of $M_1$, $M_2$, $BS_1$, and $BS_2$ (non-polarizing), is used to realize the coaxial interference of two vortex beams. The dove prism and SPPs are inserted in the MZI to generate vortex beams with different TCs required. The beam expander composed of two lenses $f_3$ and $f_4$ is inserted in one arm of the MZI. Two lenses with different focal lengths of $f_1$ and $f_2$ are inserted into two different arms of MZI, respectively, so that the two interference beams possess different radii of curvature at the detected plane.

The position of mirror $M_1$, which is mounted on a translation stage, is carefully adjusted to control the relative phase difference $\delta$ between two vortex beams. Finally, the interference patterns are recorded using the CMOS camera (CinCam, CMOS-1202).

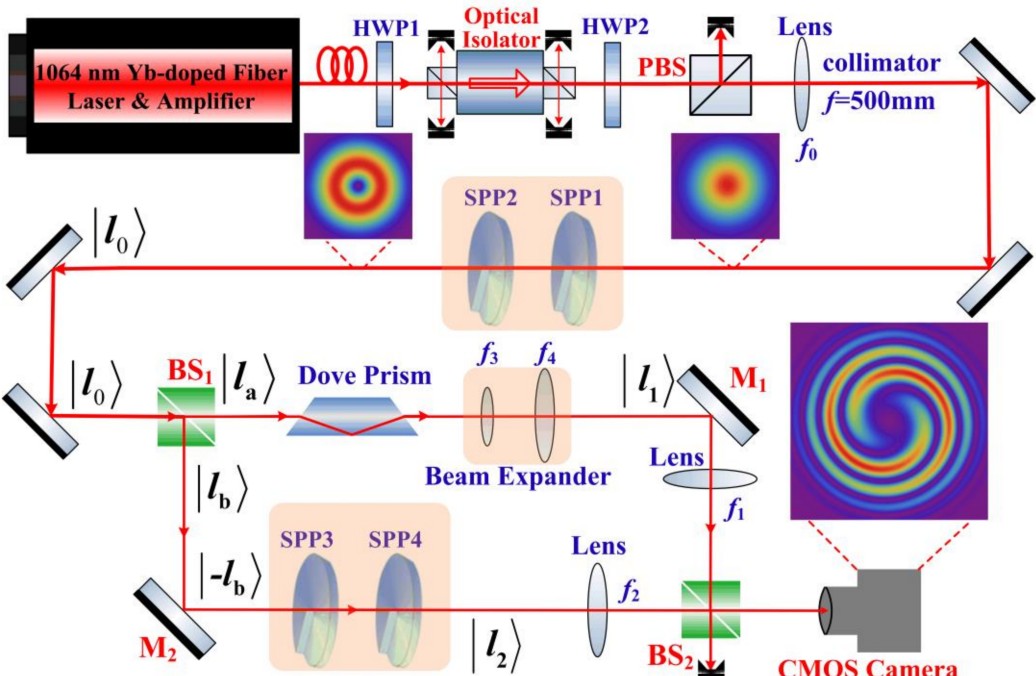

**Figure 1.** Schematic of the experimental setup; (inset); typical intensity distribution of the beams in the experiment. $M_1$ and $M_2$, high reflection mirrors; $f_0$–$f_4$, lens; HWP: half-phase plate; PBS, polarizing beam splitter; BS, 50/50 non-polarizing beam splitter.

Detail experimental procedures are as follows. (1) Although the lens $f_0$ with a focal length of 500 mm is used to collimate the laser beam output from the laser, the collimated beam is still slightly focused with beam waist of~800 μm at the position 1 m after lens $f_0$. The CMOS camera is placed at the waist position for the first time. The MZI is inserted between the lens $f_0$ and the camera. Since the dove prism and spiral phase plates are inserted into two arms of MZI, respectively, the optical paths of two arms of MZI are unequal; (2) Block the optical path of $BS_1$-$M_1$-$BS_2$, and tune the position of camera slightly for the second time to make the camera at the waist of the beam pass the $BS_1$-$M_2$-$BS_2$ path ($z_{l_2}$). Block the optical path of $BS_1$-$M_2$-$BS_2$, focus the beam passing $BS_1$-$M_1$-$BS_2$ path with the lens $f_1$ (focal length of 50 mm), and adjust the position of $f_1$ carefully to make the beam waist ($z_{l_1}$) right at the position of the camera. The waist positions of two interfering beams now coincide ($z_{l_1} = z_{l_2}$). The camera is changed for the third time to the new position, where the spot radii of two beams are the same after the position of the beam waist ($z > z_{l_1}$). Experimental results for effects of phase difference between two interference beams on spiral patterns and spiral patterns formed by coaxial interference between two vortex beams are measured; The steps (3) and (4) shown below are performed to study on effects of wavefront curvature radii relations between two interference beams on spiral patterns. (3) The beam expander is inserted in the $BS_1$-$M_1$-$BS_2$ arm of the MZI to enlarge the beam radii to twice that of its original size (the spot radii ratio of two collimated beams $w_{01}$:$w_{02}$ is 1:2). Follow the procedure shown above to coincide the waist positions of two interference beams ($z_{l_1} = z_{l_2}$) (except that the beam passing $BS_1$-$M_1$-$BS_2$ is focused by lens $f_1$ with a focal length of 200 mm). The camera is detected at different propagation distance $z$ to record the interference pattern; (4) Two collimated beams with different ratios of spot radii $w_{01}$:$w_{02}$ = 1:2 (2:1) are focused with lenses $f_1$ and $f_2$, respectively, so that different

relations for radii of wavefront curvature between two interference beams are realized at the measurement position, where the spot radii of two beams are the same.

## 4. Results and Discussion

The coaxial interference patterns of two beams under different experimental conditions (such as the effective phase difference $\delta'$, different radii of wavefront curvature $R_l(z)$, and TCs) are investigated experimentally and theoretically. The results will be presented one by one below.

### 4.1. Effects of Phase Difference between Two Interference Beams on Spiral Patterns

Effect of effective phase difference $\delta'$ between a TEM$_{00}$ Gaussian beam and a vortex beam carrying TC of $l_2 = 1$ on the lobe direction in the spiral pattern is studied and shown in Figure 2a,b. Figure 2c presents the beam radii of two interference beams versus propagation distance $z$. Since the interferograms are detected at the position where the beam spots of two interference beams are the same, the radii of wavefront curvature satisfy the relation: $1/R_{l_1} > 1/R_{l_2}$. The origination or vanishing of one interference fringe takes place at the center (exhibiting one lobe), and the twist direction rotates counterclockwise. With a gradual increase of the effective phase difference $\delta'$ from 0 to $3\pi/2$ ($2\pi$) (corresponding to Figure 2(a1–a4)) respectively, the lobe direction rotates 270° (a circle) clockwise around the core of the vortex beam.

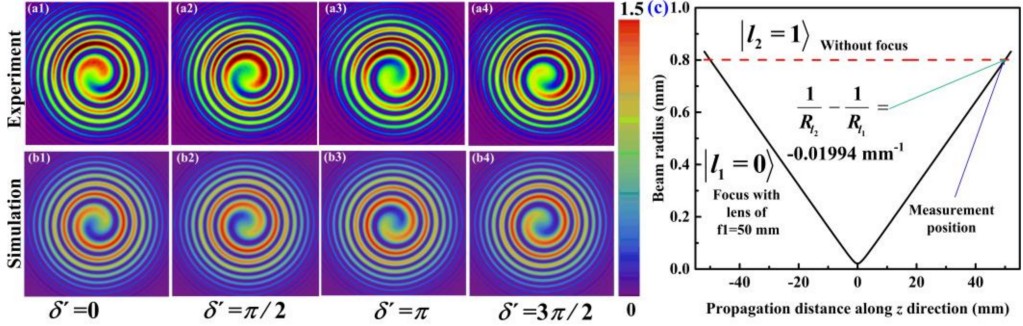

**Figure 2.** Effect of the effective phase difference $\delta'$ between the Gaussian beam ($|l_1 = 0\rangle$) and the vortex beam ($|l_2 = 1\rangle$) on the lobe direction. Images (**a**,**b**) show interference patterns; Row (**a**) shows the experimental results, and row (**b**) shows the simulation results. The color bar is shown for numerical simulation. (**c**) shows the beam radii of two interference beams versus the propagation distance.

From the expression of the effective phase difference $\delta'$, we can infer that $\delta'$ affects both the direction of the spiral lobes and the fringe brightness near the core of the beam. In order to illustrate the effect of $\delta'$ on the fringe brightness at the beam center, the interference pattern between two TEM$_{00}$ Gaussian beams are considered. The relationship of their radii of wavefront curvature is similar to that shown in Figure 2c ($1/R_{l_1} > 1/R_{l_2}$). Figure 3 shows the change of interference light intensity at the beam center versus the effective phase difference $\delta'$. Coaxial interference between two TEM$_{00}$ gaussian beams with different radii of wavefront curvature results in Fresnel rings (concentric rings). When $\delta'$ increases from 0 to $\pi$, and then to $7\pi/4$, (corresponding to Figure 3(a1–a6)), the brightness at the beam center changes from the brightest to the darkest and then gradually brighter. The running out of the rings from the center (or running them to the center), as $\delta'$ changes by $2\pi$. This is similar to Newton's rings.

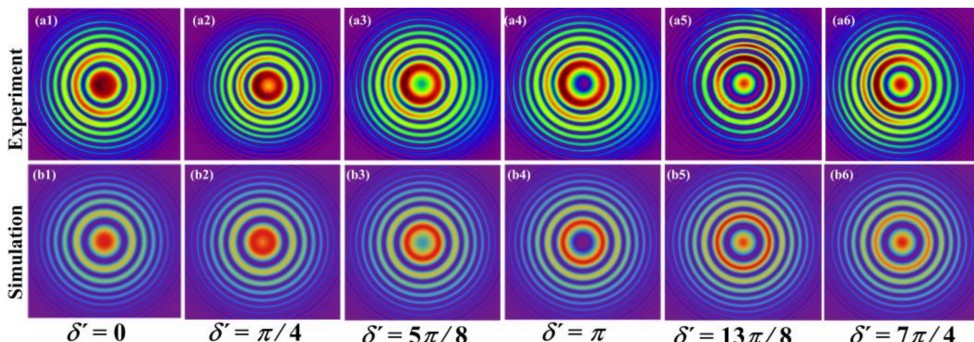

**Figure 3.** Effect of the effective phase difference $\delta'$ between two Gaussian beams ($|l_1 = 0\rangle$) on the interference light intensity at the beam center. Row (**a**) shows the experimental results, and row (**b**) shows the numerical simulation results.

*4.2. Effects of Wavefront Curvature Radii Relations between Two Interference Beams on Spiral Patterns*

Coaxial interference characteristics between a TEM$_{00}$ Gaussian beam and a vortex beam ($|l = \pm 2\rangle$) with different radii of wavefront curvature are studied in this part.

We consider the interference between a focused TEM$_{00}$ Gaussian beam (focusing by a lens $f_1$ with a focal length of 200 mm) and a collimated vortex beam ($|l_2 = 2\rangle$) first. The camera is detected at a different propagation distance $z$ to record the interference pattern under the different differences of reciprocals for wavefront curvature radii. Figure 4a,b shows the spiral pattern measured by camera and the simulated results. The beam radii of two interference beams at different propagation distance $z$ are shown in Figure 4c. As shown in Figure 4a,b, the interference pattern exhibits a spiral pattern with two lobes, and the experimentally recorded patterns are in good agreement with the theoretical expected profiles. Taking $z = 0$ as the reference position of the beam waist ($z_{l_1} = z_{l_2} = 0$). At the focusing plane ($z = 0$), we can observe a spot image as shown in Figure 4 (a5,b0) because the intensity of the focal spot is much larger than the intensity of the vortex beam. For the image plane before the focusing plane ($z < 0$), the twist direction for the spiral fringe for interference between the helical wavefront and the convergence wavefront comes into clockwise, while the twist direction of spiral fringes for interference between the divergence TEM$_{00}$ Gaussian beam and the collimated vortex Gaussian beam ($z > 0$) is opposite to the case of convergence wavefront interference. This is because the relation between the radii of wavefront curve for two interference beams changes from $1/R_{l_1} < 1/R_{l_2}$ to $1/R_{l_1} > 1/R_{l_2}$ for the propagation distance $z$ changing from negative to positive (as shown by the black solid line in Figure 4d). Meanwhile, we also notice that the direction of the lobes near the core of the pattern changes with the propagation distance $z$. This is due to the Gouy phase shifts [43,44] for two interference beams undergoing different degrees of change with $z$, thus resulting in the effective phase difference $\delta'$ changing with the propagation distance $z$ (as shown by the red dash line in Figure 4d).

We then consider the interference between a TEM$_{00}$ Gaussian beam and a vortex beam with different wavefront curvature radii. The CMOS camera is detected at the position where two interference beams have the same spot radius. Figure 5 presents the spiral patterns measured by camera and the corresponding simulations. The interference pattern between a TEM$_{00}$ Gaussian beam and a vortex beam of $|l_2 = \pm 2\rangle$ exhibits a spiral pattern with two lobes. When the interference occurs between $|l_1 = 0\rangle$ and $|l_2 = -2\rangle$, the twist direction is clockwise for $1/R_{l_1} > 1/R_{l_2}$, and counterclockwise for $1/R_{l_1} < 1/R_{l_2}$. When the interference occurs between $|l_1 = 0\rangle$ and $|l_2 = 2\rangle$, the twist direction is clockwise for $1/R_{l_1} < 1/R_{l_2}$, and counterclockwise for $1/R_{l_1} > 1/R_{l_2}$. Although the sign of vortex TCs in Figure 5(a2,a5), and in Figure 5(a3,a4) are opposite, the interference patterns are similar and the twist direction is counterclockwise. Thus, for a correct determination of the TC, it is necessary to know the relation of the wavefront curvature radii between two beams.

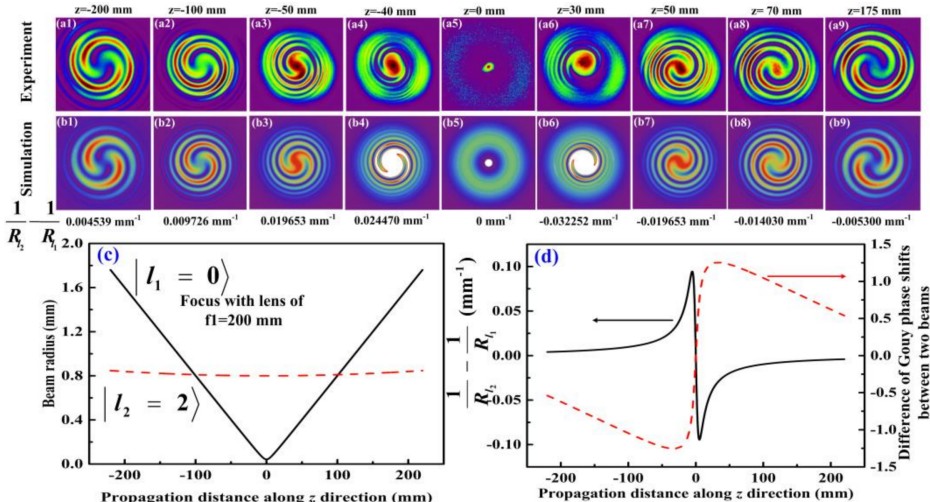

**Figure 4.** Coaxial interference characteristics between a focused TEM$_{00}$ Gaussian beam and a collimated vortex beam ($|l_2=2\rangle$). Images (**a1**–**a9**) show the experimentally measured spiral pattern, and images (**b1**–**b9**) correspond to the numerical simulations. (**c**,**d**) are the calculated spot radii and wavefront curvature radii (difference of Gouy phase shifts) of two interference beams versus the propagation distance $z$, respectively.

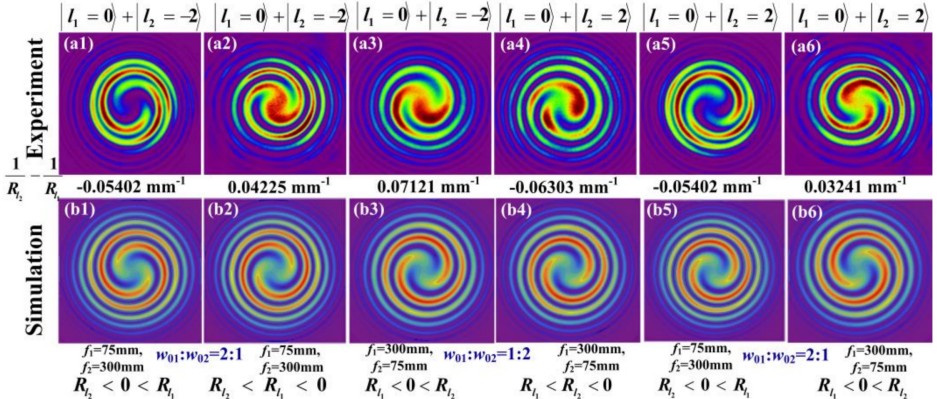

**Figure 5.** Interferograms of a focused TEM$_{00}$ Gaussian beam and a focused vortex beam ($|l_2 = \pm 2\rangle$) with different wavefront curvature radii; (**a1**–**a6**) the experimentally measured spiral pattern; (**b1**–**b6**) the numerical simulations.

### 4.3. Spiral Patterns Formed by Coaxial Interference between Two Vortex Beams

In this part, the interference characteristics for two vortex beams are studied. The interference between two vortex beams with a TCs' difference of 0 and 2 is considered first. The camera is detected at the position where the spot radius of parallel vortex beam $|l_2\rangle$ is the same as that of the divergent vortex beam $|l_1\rangle$, so that the relation: $1/R_{l_1} > 1/R_{l_2}$ holds. The interference patterns are shown in Figure 6. As shown by Figure 6a–c, the interference pattern for TCs' difference of 0 exhibits concentric rings, and the radius of the inner ring increases with the increase of the TCs value. As shown by Figure 6d–g, the interference pattern for the module of TCs' difference of 2 exhibits a spiral pattern with two lobes, and the twist direction is clockwise. The interference patterns are similar for the fixed TC's difference, except a slight difference of fringes near the center position. Numerical results agree well with the experimental ones.

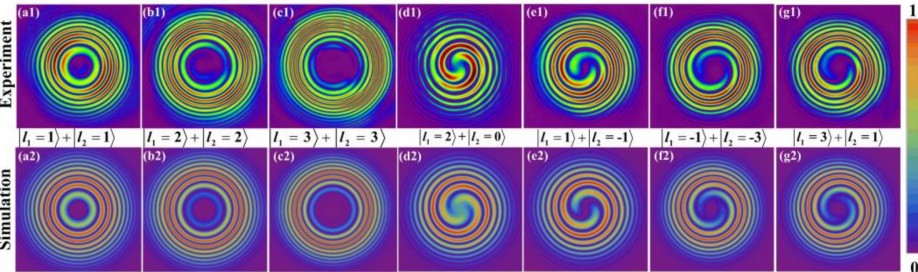

**Figure 6.** Coaxial interference characteristics between two vortex beams with TCs' difference of 0 (columns (**a**–**c**)) and 2 (columns (**d**–**g**)). Row 1 shows the experimentally measured interference patterns, and row 2 corresponds to the simulation ones.

Comparing different images shown above, we can draw the following conclusions: (i) The spiral lobes $N$ equals to the module of the TCs' difference ($N = |l_1 - l_2|$); (ii) The twist direction (clockwise or counterclockwise) depends on both the sign of $l_1 - l_2$ and $1/R_{l_1} - 1/R_{l_2}$, clockwise for the same sign, and vice versa. (iii) The spiral pattern disappears and becomes to concentric rings, when the TCs of two interference beams are the same.

Interference pattern formed by coaxial interference between two vortex beams carrying a large number of TCs is also studied numerically. Due to lack of SPPs with TCs of large value, we did not perform the experiment in this part. During the calculations below, we suppose both beams have a waist at $z = 0$, and choose $w_1 = 1$ mm, and $w_2 = 0.8$ mm, respectively. The interference pattern is detected at the position where two interference beams have the same beam spots at $z > 0$, thus resulting in $1/R_{l_1} < 1/R_{l_2}$. During the simulations, $l_2$ is kept fixed at 100, and $l_1$ is changed from 0 to 200. As shown in Figure 7, the spiral lobe number is equal to $|l_1 - l_2|$, the twist direction is clockwise for $l_1 < l_2$, and counterclockwise for $l_1 > l_2$. Although the TCs of vortex beam is as high as 100, the conclusions mentioned above still hold. The interference pattern for coaxial superposition between a Gaussian beam and a vortex beam with large value TCs is shown in Figure 7a1, the interference fringes show exactly 100 clockwise spiral fringes from the singularity. The spiral lobes are so dense that it is difficult to count them clearly. It also shows that the traditional method of coaxial interference between a Gaussian beam and a vortex beam is not suitable for measuring the large value TCs carried by the vortex beam. As shown in Figure 7a2–e2, the spiral pattern will be clear to distinguish, if the TCs of the two vortex beams are close to each other. These results indicate that the vortex beam of unknown TCs can be detected by coaxial interfering it with a reference vortex beam of known TCs.

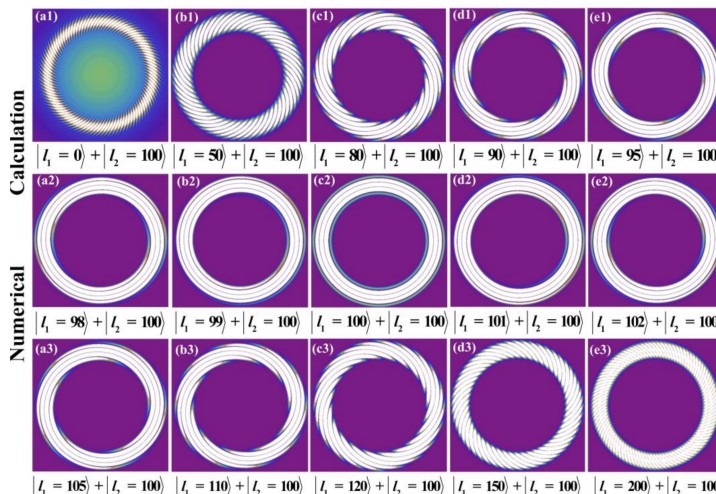

**Figure 7.** Numerical calculations of coaxial interference patterns between two vortex beams carrying TCs of large number. Row 1 shows the interference patterns between $|l_1 = 0, 50, 80, 90, 95\rangle$ and $|l_2 = 100\rangle$, row 2 shows the interference patterns between $|l_1 = 98, 99, 100, 101, 102\rangle$ and $|l_2 = 100\rangle$, and row 3 shows the interference patterns between $|l_1 = 105, 110, 120, 150, 200\rangle$ and $|l_2 = 100\rangle$.

## 5. Conclusions

In summary, a systematic study of coaxial interference patterns between two vortex beams was carried out. Three important conclusions were obtained. (i) The phase difference between two interference beams affects both the direction of the spiral lobes and the fringe brightness at the beam center; (ii) For coaxial interference between a plane wave and a spherical wave, when the spherical wave changes from convergence to divergence, the twist direction of spiral pattern will be reversed; (iii) For spiral pattern, the number of spiral lobes is determined by the absolute value of TCs' difference between two vortex beams ($|l_1 - l_2|$), while the twist direction depends on the sign of TCs' difference ($l_1 - l_2$) and difference of reciprocals for wavefront curvature radii ($1/R_{l_1} - 1/R_{l_2}$), clockwise for the same sign, and counterclockwise for opposite signs. The results deepen the understanding of coaxial interference between two vortex beams. The spiral pattern can be used in a variety of fields, such as detecting the sign and value of the TCs carried by an unknown vortex beam, measuring the curvature radius (or focal length) of the concave mirror (lens) and fabricating the chiral microstructures of materials.

**Author Contributions:** Conceptualization, J.M., P.L. and Y.G.; investigation, J.M. and P.L.; data curation, P.L., writing—original draft preparation, J.M. and P.L.; writing—review and editing, P.L. and Y.G.; funding acquisition, P.L. and Y.G. All authors have read and agreed to the published version of the manuscript.

**Funding:** This work was supported by the Scientific and Technological Developing Scheme of Henan Province (202300410072), Key Scientific Research Project of Colleges and Universities in Henan Province (18A140002), the National Natural Science Foundation of China (NSFC) (61875053, 61805068, 12004101, 12104131).

**Conflicts of Interest:** The authors declare no conflict of interest.

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
