# Peer review of "Characteristics of Spiral Patterns Formed by Coaxial Interference between Two Vortex Beams with Different Radii of Wavefront Curvatures"

_photonics, doi:10.3390/photonics8090393_

Round 1

Reviewer 1 Report

The authors of this work considered an interesting problem associated with the interference of two laser beams. One of them was an ordinary TEM00 Gaussian beam, and the second was a vortex beam with one or another topological charge and a phase delay relative to the Gaussian beam. In addition, the interference of two vortex beams with different values of the radii of curvature of the wavefront and with different topological charges was investigated. The research was carried out at a very good theoretical and experimental level. Important conclusions were drawn about the number of spiral lobs, the twist direction of the total beam depending on the sign of the topological charge difference and difference of reciprocal for wavefront curvature radius. All of this can have important practical applications. I believe that the work can be published in Photonics without changes.

Author Response

We sincerely thank you for your recognition and high evaluation of our work in this paper. We are very encouraged by your comments. We will continue to do better work. We thank you again for your careful reading of the manuscript, and good comments to the paper.

Reviewer 2 Report

The paper studies interference patterns between two Laguerre-Gauss (LG) vortex beams in dependence of their topological charges (TC) and wavefront radius of curvature.

First an expression for their interference pattern is written down and simple rules for the spiral patterns are derived, for the number of lobes and the direction of spiraling. These depend on the difference of TC and differential inverse curvature radii. Then experiments are shown that confirm these theoretical expressions.

The abstract claims that a “detailed theoretical model is established”. This I find rather overstated. The expressions for LG modes are well known and combining two of them in an interference pattern does not qualify as a detailed theoretical method.

The theory section basically gives an expression for the interference patterns to be expected, which is quite straightforward, Unfortunately it is not  written in a clear way and it makes me wonder if it corresponds to the situation studied in the experiment. The expressions for the LG beams assume that the minimum waist is always located in z=0. This is apparent from the Gouy phases [arctan(z/zR)] in Eqs. (2) and (6), as well as from the expressions for w(z) [line 118] and R(z) [line 122]. However it was not clear to me whether in the experiment the waist positions of the two interfering beams are made to coincide (in z=0). This would require careful positioning of the lenses f1 and f2, so that their focal positions coincide. I did not see this mentioned in the  text. If the waists do not coincide, I would not expect the simulations to agree with the experiment because they would refer to different situations.

This confusion should be cleared up.

There is also some confusion between z_0 and z_R. In line 118: z_0 plays the role of a Rayleigh length. This may be a typo because z_0 is later defined as  the position of the camera.

In the end, the results are just what one would expect to come out, nothing unexpected appears in my opinion. Everything was known previously, I would say, and I don’t think we learn any new physics here.  One might say that the authors provide something like a “catalog”, or overview of interference patterns for a variety of parameters.

Apart from that the results are probably correct. The theoretical description is  subject to the caveat that I described above.

The paper scores low on novelty, so I would not give it high priority for publishing. Ultimately, it is up to the editors to decide if they want to publish this type of work.

Round 2

Reviewer 2 Report

The authors have taken my comments in consideration in an appropriate manner. The paper appears to be technically correct. My opinion remains that the paper scores low on novelty.
